# The Discourse Profile in Corticobasal Syndrome: A Comprehensive Clinical and Biomarker Approach

**DOI:** 10.3390/brainsci12121705

**Published:** 2022-12-12

**Authors:** Isabel Junqueira de Almeida, Marcela Lima Silagi, Maria Teresa Carthery-Goulart, Jacy Bezerra Parmera, Mario Amore Cecchini, Artur Martins Coutinho, Sonia Maria Dozzi Brucki, Ricardo Nitrini, Eliane Schochat

**Affiliations:** 1Department of Physical Therapy, Speech, and Occupational Therapy, Hospital das Clínicas, Faculdade de Medicina da Universidade de São Paulo (HC-FMUSP), São Paulo 05360-160, Brazil; 2Cognitive and Behavioral Neurology Research Group, Department of Neurology, University of São Paulo, São Paulo 01246-903, Brazil; 3Department of Speech, Language and Hearing Sciences, Universidade Federal de São Paulo, São Paulo 04023-062, Brazil; 4Mathematics, Computing and Cognition Center (CMCC), Federal University of ABC (UFABC), Santo André 09210-580, Brazil; 5INCT-ECCE (Instituto Nacional de Ciência e Tecnologia sobre Comportamento, Cognição e Ensino), São Carlos 13565-905, Brazil; 6Department of Neurology, Hospital das Clínicas, Faculdade de Medicina da Universidade de São Paulo (HC-FMUSP), São Paulo 01246-903, Brazil; 7Human Cognitive Neuroscience, Psychology Department, University of Edinburgh, 7 George Square, Edinburgh EH8 9JZ, UK; 8Laboratory of Nuclear Medicine (LIM-43), Nuclear Medicine Center and Division, Hospital das Clínicas, Faculdade de Medicina da Universidade de São Paulo (HC-FMUSP), São Paulo 01246-903, Brazil

**Keywords:** corticobasal syndrome, corticobasal degeneration, connected speech, language, discourse, spontaneous speech, positron emission tomography, amyloid-PET

## Abstract

The aim of this study was to characterize the oral discourse of CBS patients and to verify whether measures obtained during a semi-spontaneous speech production could differentiate CBS patients from controls. A second goal was to compare the performance of patients with CBS probably due to Alzheimer’s disease (CBS-AD) pathology and CBS not related to AD (CBS-non-AD) in the same measures, based on the brain metabolic status (FDG-PET) and in the presence of amyloid deposition (amyloid-PET). Results showed that CBS patients were significantly different from controls in speech rate, lexical level, informativeness, and syntactic complexity. Discursive measures did not differentiate CBS-AD from CBS-non-AD. However, CBS-AD displayed more lexical-semantic impairments than controls, a profile that is frequently reported in patients with clinical AD and the logopenic variant of primary progressive aphasia (lvPPA). CBS-non-AD presented mainly with impairments related to motor speech disorders and syntactic complexity, as seen in the non-fluent variant of PPA.

## 1. Introduction

Corticobasal syndrome (CBS) is a rare neurodegenerative syndrome, within the group of atypical parkinsonian disorders. It is characterized by motor and higher cortical function impairments [1,2,3,4]. CBS was once thought to be a unique clinicopathological entity related only to corticobasal degeneration (CBD) pathology, but currently, it is well known that CBS may be the clinical manifestation of a variety of underlying pathologies, including tauopathies, as well as Alzheimer’s disease (AD) [4,5,6]. Due to this heterogeneity, many studies have been trying to identify clinical features and biomarkers capable of predicting the underlying pathology of CBS [7,8,9,10,11,12].

Regarding the clinical characteristics of CBS, motor symptoms usually occur asymmetrically and include dystonia, myoclonus, and akinetic-rigid parkinsonism. Higher cortical dysfunctions include apraxia, alien limb phenomena, cortical sensory loss, behavioral changes, and cognitive impairment [1,5]. There is a great heterogeneity of cognitive symptoms which may impact executive functions, memory, visuospatial abilities, social cognition, and language [13].

CBS may cause impairments in all language processing levels, and previous research could not identify a unique pattern of change [14,15]. Most studies reported a phenotype similar to the non-fluent variant of primary progressive aphasia (nfvPPA) [16,17,18], whereas others reported a wide variety of phenotypes: Broca’s aphasia, anomic aphasia, fluent aphasia, mixed aphasia and logopenic variant of PPA (lvPPA) [2,10,15,19,20]. The presence of motor speech disorders (dysarthria and apraxia of speech) has also been reported [21,22].

Language and motor speech features have been investigated as predictors of underlying pathology in CBS [7,8,9,10,12]. For instance, impaired repetition has been associated with biomarkers of AD pathology [7,11] and apraxia of speech with tauopathies underlying pathologies [7]. It is notable that some CBS patients present with bilateral parietal hypometabolism, more marked in the left inferior parietal gyrus [11], which is one of the imaging characteristics that support the diagnosis of the lvPPA, an atypical presentation of AD. In line with these findings, a retrospective study conducted with a large cohort of CBS patients revealed that those with AD confirmed pathology presented with anomia, word retrieval difficulties, and poor sentence repetition [10]. The authors suggested that these symptoms (i.e., a lvPPA phenotype) are helpful to predict AD underlying pathology among CBS patients [10]. On the other hand, other studies identified more severe language disorders among CBS patients without AD biomarkers. These patients presented lower performances in picture naming and word comprehension [8] and/or the clinical profile of nfvPPA [11]. Motor speech symptoms have also been proposed as possible predictors of non-AD underlying pathology in CBS. Parjane and colleagues found an association between CSF pTau levels and limited pitch range of the fundamental frequency [12]. Our group conducted a prospective study with 31 CBS patients and evidenced that dysarthria was significantly more frequent among amyloid-negative CBS patients [9].

### 1.1. Assessment of Oral Discourse Production in CBS

Oral discourse, i.e., an extended language production [23], is considered the basis for everyday communication [24], and an essential part of language assessment [23,25,26,27,28]. Differences in connected speech measures have been evidenced between typical and pathological cognitive aging, and between distinct neurodegenerative syndromes [25]. However, most studies with CBS patients have not included discourse analysis, restricting their assessment to decontextualized language production tasks, such as confrontation naming, verbal fluency, and repetition [14,25].

Analyzing connected speech samples is a simple and ecological form of conducting a comprehensive assessment of cognitive and linguistic aspects of communication [25,29,30]. Research on discourse production in adults has focused on three aspects [24]: how *language* is used for discourse production; what information is conveyed in discourse (*coherence*); and how this information is structured (*cohesion*). Regarding language, connected speech measures can inform about dysfunctions on phonetic-phonological, lexical-semantic and morphosyntactic levels. Regarding local and global coherence, connected speech analysis focuses on the number and relevance of correct information units (CIU) and on how information is logically integrated into the narrative [25,26,31]. Finally, concerning structure, connected measures inform about the ability to link and reference elements within and across sentences to reduce discourse ambiguity. Speech rate and fluency are also usually investigated through connected speech samples [25,32,33].

The most utilized forms to elicit connected speech are narrative production, picture description, and conversation. The first two are considered semi-spontaneous speech tasks, as there are restrictions imposed by the picture or the story/topic proposed by the examiner [23,25]. Picture description tasks present the benefits of reducing demands on memory and attention, as the picture is available to the patient during the whole task, and it facilitates the comparison across subjects, as the description is expected to present similar key elements. On the other hand, this task usually requires a limited variety of syntactic structures, mostly eliciting sentences in the present tense [23,25,26].

Connected speech remains poorly investigated in CBS patients. To the best of our knowledge, only two studies analyzed discursive measures in these patients [12,34].

One study evaluated 20 patients with CBS, using a story-telling task based on a wordless book [34]. The authors focused on narrative organization and coherence. They found deficits in three variables: narrative theme (the ability to maintain the topic of the story), global coherence (identification of the main point of the story), and local coherence (ability to link the consecutive events of the story). The authors demonstrated that CBS patients could correctly identify the elements in the picture, but failed to interpret how they could be integrated to build a coherent narrative [34].

Parjane and colleagues [12] used an automated approach to analyze acoustic and lexical measures extracted from a semi-spontaneous speech sample. They compared three groups: CBS and progressive supranuclear palsy (PSP), nfvPPA and healthy controls. CBS/PSP and nfvPPA patients presented similar speech features that were different from those obtained in the control group. Both patient groups presented a reduction in the pitch range of the fundamental frequency, shorter speech segments, longer and more frequent pauses, and a reduced speech rate, which was associated with the presence of grammatical impairment. The authors discuss that their findings reinforce the idea that CBS/PSP and nfvPPA are part of the same spectrum [12]. However, nfvPPA patients were more impaired than CBS/PSP patients in the duration of speech segments, frequency of pauses, speech rate (slower), and verb production (reduced).

As mentioned previously, CBS can be the phenotype of different neurodegenerative diseases, including tauopathies and AD. Connected speech has more often been investigated in PPA (that is frequently caused by tauopathies and AD) and clinical AD; therefore, in the following sections, we review the major findings of these studies.

### 1.2. Connected Speech Analysis in Primary Progressive Aphasia

A review on connected speech in neurodegenerative disorders discussed 15 studies involving PPA. The majority investigated nfvPPA and the semantic variant of PPA (svPPA) [25]. Compared to healthy controls, nfvPPA patients present a lower speech rate, phonemic and phonetic errors [25,32,35,36,37]. In the study conducted by Wilson and colleagues [32], phonemic errors were produced by some of the nfvPPA patients, while distortions were evidenced in all cases. The lexical level is relatively spared in this variant [25,32]. Patients with nfvPPA may produce more content than function words [32,38] andfewer adjectives and adverbs compared to lvPPA and svPPA [38].

At the syntactic level, nfvPPA patients have difficulty generating complex syntactic structures, with a reduced mean length of utterance and reduced number of embeddings, but not with abundant morphosyntactic errors [25,32]. Incomplete sentences are also frequently reported in this variant [37]. These findings suggest that most patients are not simply agrammatic, as patients with Broca’s aphasia. Graham and colleagues [39] analyzed connected speech measures in a group of PPA (9 nfvPPA, 14 svPPA, and 4 lvPPA). Speech was evaluated by experienced raters, according to a checklist of symptoms for frank agrammatism. Only two patients were rated as frank agrammatical, and both were nfvPPA patients.

In svPPA, the speech rate seems to be slightly reduced in comparison to healthy control subjects [25,27,32]. Studies do not report phonetic or phonemic errors [25,32,35]. The number of content words is reduced [25,32,35], especially for nouns [32,35,36,38], which is in accordance with their difficulty in confrontation naming tasks [40]. Semantic errors are frequent in this group [25,27]. The syntactic level seems to be spared, with few errors being reported. These errors are rather paragrammatic than agrammatic [32].

The connected speech of lvPPA patients is marked by a low speech rate, and disfluencies, such as filled pauses, false starts and repaired sequences [25,32,35]. Compared to the other PPA variants, the speech rate was classified as intermediate, i.e., slower than svPPA and faster than nfvPPA [32]. Phonemic paraphasias are present, while distortions rarely occur [25,32]. The number of open-class words may be reduced, while the proportion of pronouns is increased [25,32,35,41]. This finding may be related to anomia, as the lvPPA patients tend to replace nouns for pronouns [41]. Some studies report a higher proportion of verbs than nouns [32,41]. Like patients with semantic variant, syntactic errors are rather paragrammatic than agrammatic [25,32]. This difficulty is explained by working memory deficits, leading to a failure in processing syntactic dependencies [32], and/or word-finding difficulties, resulting in incomplete sentences [41].

### 1.3. Connected Speech Analysis in Alzheimer’s Disease

On the phonetic-phonological level, patients diagnosed with AD have a slower speech rate and hesitations on connected speech tasks [23,25,27]. Phonemic errors have been reported in mild AD patients [27]. On the lexical-semantic level, AD patients produce a greater number of closed-class words [25,27,29] and pronouns, more high-frequency words and have more word-finding episodes, word repetitions, and revisions [23,25,26]. Regarding the syntactic level, AD patients tend to use simple syntactic structures, such as reduced sentences and short utterances [23,25,26]. Inflectional errors are reported in some studies [25,27]. Impairments on discourse and pragmatic levels are often reported [25]. AD patients produce fewer information units, which refers to a piece of correct and nonredundant information about the picture [23]. Their speech is less efficient, which means that they need more time or a greater number of words to convey information [23,25].

### 1.4. The Present Study

The characterization of discourse production is relevant both to support the clinical diagnosis of CBS, as well as to design behavioral interventions to improve functional communication and the quality of life of these patients. Furthermore, with the development of new disease-modifying therapies for AD and tauopathies, it becomes increasingly necessary to recognize the clinical characteristics of each pathological condition [10,13,35,42].

The current exploratory study aims to shed light on this topic. Our main objective is to characterize the oral discourse of a sample of 14 CBS patients, by analyzing discursive measures of all language levels: phonetic-phonological, lexical-semantic and morphosyntactic. Additionally, we aim at investigating whether discursive measures can contribute on the differentiation between CBS patients with and without AD underlying pathology, based on a brain metabolic dichotomized analysis using positron emission tomography with [^18^F]fluorodeoxyglucose (FDG-PET) and in the presence or absence of amyloid deposition using amyloid-PET. Previous studies have described the heterogeneity of language disorders in CBS patients [14,15]; therefore, we hypothesize that, as a group, CBS patients will present deficits in all language levels. Concerning the differentiation of CBS clinical profiles associated or not with AD underlying pathology, based on previous studies (e.g., [11]), we hypothesize that CBS patients with AD pathology will present a clinical profile similar to the lvPPA, with more lexical retrieval deficits, while CBS not related to AD pathology (i.e., tauopathology) will have a clinical profile similar to the nfvPPA, with more motor speech and morphosyntactic deficits.

## 2. Methods

### 2.1. Participants

Fourteen patients (9 female; mean age 67.9; 8.8 ages of education) with a clinical diagnosis of probable CBS were prospectively recruited at the Movement Disorders and Behavioral Neurology Units of Hospital das Clínicas (University of São Paulo, Medical School, São Paulo, Brazil), between February 2017 and November 2019. The clinical diagnosis of CBS was made by two experienced neurologists (S.M.D.B. and J.B.P.), according to current criteria [1].

The CBS group was further divided into CBS-non-AD (CBS likely not related to AD), and CBS-AD (CBS likely related to AD), according to the presence or absence of cortical amyloid deposition on the [^11^C]Pittsburgh Compound-B positron emission tomography (PIB-PET) and the FDG-PET metabolic patterns, as will be detailed in Section 2.4. A healthy control group (CG) with 15 volunteers, matched by age and education with the patients, was evaluated between October 2018 and June 2020.

Inclusion criteria for the CBS group were: Brazilian Portuguese as native language, being able to produce an oral narrative, and speech production sufficiently intelligible, i.e., the majority of words could be understood and transcribed. Exclusion criteria were: relevant nondegenerative structural lesions (e.g., stroke and tumors) and a well-defined premorbid psychiatric disease. Inclusion criteria for CG were: Brazilian Portuguese as native language; absence of cognitive complaints; no history of neurological or psychiatric diseases; not being medicated with drugs that could affect cognition and a normal range score in the Addenbrooke’s Cognitive Examination-revised (ACE-R) adjusted for age and education [43,44].

This study was approved by the Ethics Committee of the Hospital das Clínicas da Universidade de Sao Paulo (CAPPesq) (protocol code 02874318.9.0000.0068, 4 May 2017). All participants or their caregivers gave written informed consent to participate in this study.

### 2.2. Clinical and Neuropsychological Assessment

The CBS group underwent a comprehensive neurological examination, a brief cognitive assessment, and a comprehensive language evaluation. The CG group underwent the same cognitive and language tests. All tests were adapted to Brazilian Portuguese, according to the cutoff scores for the Brazilian population.

The neurological examination comprised the evaluation of motor signs such as parkinsonism, dystonia, myoclonus, pyramidal signs, postural instability, tremor, and ocular motor dysfunction. Functional motor impairment was categorized by the Hoehn and Yahr scale (H&Y) [45]. Limb and orobuccal apraxia were verified by imitation of gestures with and without meaning and imaginary tool use [46]. The Functional Activities Questionnaire (FAQ) [47,48] was used to investigate functional decline. Dementia staging was classified according to the Clinical Dementia Rating scale (CDR) [49,50]. The presence of alien limb phenomena, visual neglect, cortical sensory loss, and Balint and Gerstmann syndromes signaled cortical signs. The Neuropsychiatric Inventory (NPI) [51,52] was used to identify behavioral changes.

Global cognition was evaluated through the ACE-R [43,44], a battery that includes subtests of attention, memory, verbal fluency, language, and visual-spatial abilities, and a total score.

Language abilities were assessed using the Western Aphasia Battery—Revised (WAB-R) [53,54]. The following WAB-R tests were utilized: spontaneous speech (conversational questions and picture description), auditory verbal comprehension (yes/no questions, auditory word recognition and sequential commands), repetition, and naming and word finding (object naming, word fluency, sentence completion, and responsive speech). Based on those tasks, the Aphasia Quotient (AQ) was derived, which is a measure of aphasia severity.

### 2.3. Discourse Assessment

The “Picnic Scene” picture description task from WAB-R [53] was utilized to elicit the oral discourse. The subjects were instructed to look carefully at the picture and to describe it using sentences without a time limit. Patients were encouraged to use sentences instead of single words. All samples were video recorded. The quantitative procedure of transcription and analysis was adapted from other studies on neurodegenerative disorders [32,33].

One author (I.J.d.A.), with experience in transcription, manually transcribed all samples in an orthographic manner. Each transcription was checked by a second author (M.L.S.), using the original videos. Disagreements were discussed until a consensus was reached. Non-narrative words, such as coordinating conjunctions, questions addressed to the examiner, and direct answers to a question from the examiner were transcribed but not analyzed. Each speech sample was divided into utterances, i.e., sequences of words not interrupted by pauses and lasting more than two seconds. An utterance could be a single word, a nominal or verbal phrase, or a complete sentence. Conservative decisions were taken, leading to shorter rather than longer utterances.

The discourse features analyzed were adapted from previous clinical studies focusing on neurodegenerative disease [32,33]. Nineteen variables were established for analysis, divided into four categories: (1) speech rate and speech sound errors, (2) other disruptions to fluency, (3) lexical-semantic level, and (4) syntactic structure and complexity. CIU [55], a measure of informativeness of the discourse, was included at the lexical-semantic level. This measure was calculated according to the recommendations described by Nicholas and Brookshire [55]. All variables are described in Table 1. One author (I.J.d.A.) analyzed each transcription, and a second author (M.L.S.) checked all the information. Both authors discussed divergences until consensus.

### 2.4. Neuroimaging Examination

CBS patients underwent a multimodal neuroimaging examination, with magnetic resonance imaging (MRI), PIB-PET and FDG-PET, which is described in detail in the methods section of a previous publication from our group [42].

Both [^11^C]Pittsburgh Compound-B (PIB) and [^18^F]fluorodeoxyglucose (FDG)-positron emission tomography (PET) were produced in an on-site cyclotron (PET trace 880, GE Healthcare) at the Nuclear Medicine Center of the Institute of Radiology (CMN InRad, São Paulo, Brazil) of our Hospital. PIB-PET and MRI images were simultaneously acquired on a hybrid 3.0-Tesla SIGNA PET/MRI scanner (GE Healthcare, Milwaukee, WI, USA). FDG-PET was acquired in a Discovery 710 PET/CT scanner (GE Healthcare, Milwaukee, WI, USA).

The FDG-PET was performed within one month after clinical examination, and the time between FDG and PIB-PET varied from 2 days to 6 months. Visual analysis of FDG-PET images was assisted by the 3D-SSP semi-quantitative software (Cortex ID Suite, GE healthcare) and normalized by at least two different methods (global cortex and pons). Two board-certified nuclear physicians, blinded to each other’s interpretation, clinical profile, and PIB-PET status performed the visual analysis of PET-FDG images. Based on the FDG-PET findings, the patients were split into two groups, namely “CBS likely related to AD” (CBS FDG-AD), or “CBS likely not related to AD” (CBS FDG-nonAD). Hypometabolic patterns suggestive of AD included decreased regional brain glucose metabolism (rBGM) in the posterior temporoparietal, inferior temporal regions, precuneus, and posterior cingulate gyrus.

The same nuclear medicine physicians blindly evaluated the PIB-PET images. Participants were rated as “amyloid positive” (CBS-A+) or “amyloid negative” (CBS-A-), according to previously established criteria [56,57].

The MRI protocol included volumetric T1, T2, FLAIR sequences, and diffusion-weighted imaging in 6 and 33 directions, and susceptibility-weighted imaging. Images were visually inspected by a board-certified neuroradiologist for the detection of structural brain lesions, and artifacts that could impair imaging processing.

Thus, based on PIB-PET and FDG-PET, patients were split into two groups: CBS-AD (probable AD underlying pathology) and CBS-non-AD (other possible non-AD underlying pathologies). If the patients did not perform PIB-PET, FDG-PET was considered as a surrogate for underlying pathology to assign the patients to the groups. Previous work has shown that FDG-PET has high accuracy for predicting positive amyloid deposition and consequently identifying probable underlying AD pathology in CBS patients [9].

### 2.5. Statistical Analysis

For the descriptive analysis, the means and standard deviations were calculated for all variables. Comparisons were first made between CG and patients with CBS using the Mann–Whitney test. CBS patients were then grouped according to PIB or FDG-PET status, as either CBS-non-AD or CBS-AD and these groups were compared to each other and to CG using the Kruskal–Wallis test. Bonferroni’s correction for multiple comparisons was used for each variable individually—the adjusted *p*-value, then, was the uncorrected *p* multiplied by three. A chi-squared test was performed to compare categorical variables (sex and hand dominance). All tests were two tailed, and a significance level of 0.05 was set for all analyses. The minimum and maximum values of all variables were described in Appendix A, in addition to the performance of each patient and the statistics from the Kruskal–Wallis and chi-square tests. The Statistical Package for Social Sciences software, version 26.0 (SPSS, IBM Statistics, Chicago, IL, USA), was used for the analysis.

## 3. Results

### 3.1. Demographic and Clinical Characteristics

The groups were equivalent in sociodemographic characteristics (age, education, sex, and hand dominance), as shown in Table 2. There were also no statistically significant differences in clinical variables between CBS-non-AD and CBS-AD, considering the time of symptom onset, the severity of the disease (CDR), the degree of functionality (FAQ), neuropsychiatric (NPI), and motor symptoms (H&Y).

### 3.2. Neuroimaging

All patients underwent FDG-PET. Patients showed a predominantly asymmetrical hypometabolism comprising frontal, temporal and parietal areas, contralateral to the most affected side. Twelve patients also underwent PIB-PET examination. Two patients did not undergo PIB-PET because they were institutionalized. Regarding PIB-PET analyses, 8 of 12 patients (66.6%) had negative results and 4 out of 12 (33.3%) showed positive results. The remaining two patients who did not undergo PIB-PET had a non-AD pattern on FDG-PET. Therefore, patients were divided into CBS-AD (*n* = 4) and CBS-non-AD (*n* = 10).

### 3.3. Neuropsychological Assessment

The CBS group had lower scores in comparison to CG in all cognitive subtests (ACE-R) (Table 3). Concerning CBS subgroups, none of the subtests differentiated CBS-non-AD from CBS-AD, but both groups maintained lower scores in comparison to controls, except in the language subtest. In this measure, CBS-non-AD were significantly different from controls. On the comprehensive language evaluation, the CBS group was impaired compared to controls in all measures from WAB-R (Table 3). When divided into subgroups, CBS-AD and CBS-non-AD were significantly impaired in all WAB-R subtests in comparison to controls, except in auditory comprehension. In this subtest, only CBS-AD was different from controls (*p* = 0.044) and a marginally significant difference was found for CBS-non-AD (*p* = 0.053). No measure differentiated CBS subgroups from each other.

### 3.4. The Discourse Profile

When comparing the CBS group to CG on “speech rate and speech sound errors”, the CBS group had a lower speech production rate (number of words/duration of discourse) (Table 4). On “other disruptions to fluency”, no variable differentiated the groups (Table 4). On “lexical-semantic level”, the CBS group had a lower proportion of open-class words (open-class/closed-class words), a lower percentage of CIU, and a higher proportion of lexical-semantic errors than CG (Table 5). Among these errors, 93.33% were word-finding difficulties, and 6.6% were semantic paraphasias, the latter being produced by only two patients. On “syntactic structure and complexity”, only the proportion of embeddings differentiated CBS patients from controls (Table 6).

When comparing CBS-non-AD to CBS-AD, we found no statistical difference in the discursive measures. However, when comparing each subgroup to CG, some differences were found. CBS-non-AD had a lower speech production rate (Table 4) and a lower proportion of embeddings (Table 6), while CBS-AD had more filled pauses (Table 4) and more lexical-semantic errors (Table 5). Both CBS subgroups had a lower proportion of open-class words (Table 5).

## 4. Discussion

The main purpose of this preliminary study was to characterize the oral discourse of patients with a clinical diagnosis of CBS, considering motor speech and language features, grammar, cohesion and coherence (informativeness). Thus, we conducted group comparisons between CBS patients and healthy control subjects, in connected speech measures elicited from the “Picnic Scene” picture from the WAB-R. Our second goal was to verify whether the discursive profile would differ according to the underlying pathology of CBS. Therefore, the CBS group was subdivided into CBS-non-AD and CBS-AD according to PIB or FDG-PET status, after a brain metabolic dichotomized analysis using positron emission tomography with [^18^F]fluorodeoxyglucose (FDG-PET) and in the presence or absence of amyloid deposition using amyloid-PET. Additionally, to better characterize the sample, the groups underwent a cognitive and language assessment.

In the next paragraphs, we discuss the cognitive-linguistic performance of the full sample of patients with CBS with an emphasis on the comparison between CBS-non-AD and CBS-AD subgroups. Then, we deepen the discussion on the general characterization of the oral discourse in the CBS group and on the different profiles of discursive impairment found in CBS-non-AD and CBS-AD subgroups.

### 4.1. Cognitive-Linguistic Performance of CBS Patients

In the general cognitive and language characterization, results showed that CBS patients had lower scores in all standardized subtests of language (WAB-R) and cognition (ACE-R). Impairment affecting different cognitive domains in CBS has been previously reported [8,44,58,59]. Regarding formal language assessments, previous studies evidenced different patterns of impairment. Graham et al. [19] found predominantly phonological deficits, whereas Peterson and colleagues [60] reported motor speech, phonological, semantic and syntactic deficits, but preserved auditory-verbal working memory. Similarly to our study, Di Stefano et al. [8] showed abnormal scores in all language domains. This heterogeneous pattern of impairments may be related to different stages of the disease, different tasks (e.g., naming and comprehension of real objects vs. pictures), and/or different underlying pathologies [15].

When comparing CBS patients according to the underlying pathology, none of the standardized cognitive or language subtests differentiated the CBS subgroups. However, CBS-AD had lower scores than CBS-non-AD on attention, memory, visuospatial abilities, and the ACE-R total score, without statistical significance, possibly due to the small sample size. Similar results were reported by Burrell et al. [7], who suggested that a more severe cognitive impairment could be an indicator of underlying AD pathology. CBS-AD patients from the study of Shelley et al. [18] were also more impaired than CBS patients with CBD on ACE total score, memory and orientation subtests.

On WAB-R, the scores were similar between CBS subgroups. Few studies compared the performance on language tests according to the underlying pathology of CBS patients. Di Stefano et al. [8] found that CBS patients without AD had characteristics of svPPA, while CBS-AD patients displayed more frequently Gerstmann syndrome, a pattern that was not confirmed by our study.

### 4.2. Discursive Performance of CBS Patients

Regarding discursive measures, as a group, CBS patients presented a lower speech production rate, a lower proportion of open-class words, a higher proportion of lexical-semantic errors, a lower percentage of CIU and a lower proportion of embeddings.

Motor speech disorders (dysarthria and apraxia of speech) are frequently reported in CBS patients [15,22,61]. These impairments may explain the reduction in speech rate found in previous studies [22] and also in our sample. Distortions, another characteristic of motor speech disorders [62], were evidenced in six patients (42.8%) (Appendix A), although not reaching a significant statistical difference. Parjane and colleagues also found a reduced speech rate in CBS/PSP patients compared to control and to nfvPPA groups, which was associated with grammatical impairment [12]. Grammatical difficulties may also have influenced the speech production rate in our sample, as the proportion of embeddings was lower in the CBS group.

CBS patients produced a lower number of words (marginally significant difference), particularly for open-class words, where a significant difference was found. We hypothesize that the reduced proportion of open-class words is related to a general reduction in the language output, a characteristic that has been previously reported in CBS patients [63]. During the discursive task, CBS patients needed to be encouraged to continue their speech, resulting in numerous prompts from the examiner. As it can be noticed in the following examples, CBS patients restricted their utterances to the obligatory elements of the sentences (the verb and its arguments, and function words), while healthy subjects enriched their discourse with adjuncts, probably resulting in more open-class words. The reduction in verbal initiative, despite the preserved capacity to generate grammatically well-formed sentences, has been previously reported not only in CBS patients [63], but also in non-semantic variants of PPA [27]. This pattern is sometimes named “dynamic aphasia” [27,63] and is also similar to transcortical motor aphasia [64].

Healthy subjects: *Um barco ao longe com duas pessoas* (“A boat in the distance with two people”)/*Tem um barco à vela pra de vez em quando passear* (“there is a sailboat for an occasional trip”)/*A criança brincava na terra* (“the child was playing on the ground”).

CBS patient: *O barco vem aqui* (“The boat comes here”)/*Aqui é um piquenique* (“Here is a picnic”)/*Uma criança* (“a child”).

On the syntactic level, the proportion of embeddings was significantly lower for CBS patients; however, morphosyntactic errors were absent. This is surprising, as the current diagnostic criteria [1] recognize nfvPPA as a possible cognitive subtype of CBD, and agrammatism is one of its core features [40]. In our study, CBS patients’ speech did not present characteristics of frank agrammatism. Closed-class words and verbal inflexions were not omitted and verbal inflection errors were also absent. However, the syntactic complexity of their sentences was reduced compared to healthy controls (lower proportion of embeddings).

In addition to being a core feature of nfvPPA, studies have demonstrated that some patients with nfvPPA do not show frank agrammatism in connected speech tasks. In the study of Wilson et al. [32], most non-fluent patients presented with a reduced mean length of utterance and reduced number of embeddings, but not with abundant morphosyntactic errors. Likewise, Graham et al. [39] showed that most non-fluent patients were not frank agrammatical. The CBS patients in our study have a similar profile in connected speech: non-fluent speech, characterized by simple sentences (reduced syntactic complexity) and without morphosyntactic errors.

Although the type of task we used for eliciting discourse is known to be less sensitive to syntactic deficits [25], cognitively healthy subjects with the same educational level produced more complex structures than those observed in CBS patients.

Regarding lexical-semantic errors, the CBS group was impaired relative to controls. Word-finding difficulties were frequent, while semantic paraphasias were produced by only two patients. Thus, we suggest that the reduced percentage of CIUs in patients’ discourse is related to lexical-retrieval deficits instead of a frank semantic deterioration, in line with previous findings in CBS [15]. However, CIU was not sensitive to differentiate semantic and non-semantic PPA variants [65]. Faroqi-Shah et al. [65] found that PPA patients could be differentiated from healthy controls and people with mild cognitive impairment based on the percentage of CIU (fewer than 70% CIUs). However, this measure was not able to differentiate between PPA subtypes, as word retrieval deficits are common to the three variants.

Interestingly, phonological paraphasias were not identified in the discourse of any of the patients, indicating that phonological encoding was spared in our sample. The literature is controversial regarding this aspect. Di Stefano and colleagues [8] and Burrell and colleagues [7] found phonemic paraphasias in 52% and 46% of their sample, respectively, on spontaneous speech tasks. Catricalà and colleagues [66] studied naming errors (classified as visual, semantic, or phonological) in different neurodegenerative diseases, and found that CBS patients produced mainly semantic errors and only a few phonological errors. These discrepancies may be due to the highly heterogeneous language profile of CBS patients [15]. However, they can also be explained by the difficulty in differentiating distortions from phoneme substitutions. Although the underlying mechanisms are different (motor vs. phonological), the behavioral manifestation may be very similar. Phonological deficits are usually present in different tasks (naming, reading, writing, repetition). On the other hand, distortions are linked to motor speech control. In this study, phonological manifestations were assessed not only in the discourse task but also in a comprehensive language assessment (WAB-R), and we found that phonological encoding was preserved across all tasks.

Another hypothesis for these discrepancies is that different languages may have different manifestations in neurodegenerative diseases, a topic that is still poorly investigated in the literature on neurodegenerative diseases. Canu et al. [33] compared connected speech features of English and Italian native speakers with nfvPPA. English speakers presented with greater motor speech impairment, while Italian patients presented with greater grammatical impairment. The authors relate these deficits to the specificities of each of these languages: English with greater articulatory complexity and Italian with a more complex morphology. Thus, speech and language manifestations may vary according to the patient’s native language with calls for more cross-language comparison studies.

Finally, we cannot exclude that impairments in non-linguistic cognitive domains, such as visuospatial functions, attention, non-verbal memory, and executive functions have influenced the performance of CBS patients. Discourse production requires more than just linguistic abilities. The relationship between language and cognition has been widely investigated in mild cognitive impairment and AD [67,68]; however, it remains underexplored in other neurodegenerative diseases. CBS patients may have had difficulty in accessing, integrating, and maintaining information due to failures in different memory subsystems (working, semantic, and episodic memories), in addition to discourse disorganization resulting from executive dysfunction, impacting the micro and macrostructure of discourse (i.e., grammar cohesion and coherence). Nonetheless, the relation between cognition and discourse was not explored in this study.

### 4.3. Discourse Performance in CBS-Non-AD and CBS-AD Patients

When comparing CBS patients according to the underlying pathology on connected speech, no measures differentiated CBS-non-AD from CBS-AD. However, when the groups were compared to controls, some differences were found. CBS-non-AD presented a lower speech production rate and a lower proportion of embeddings. As discussed above, this is a speech pattern similar to nfvPPA, with motor speech disorders, but without frank agrammatism. This pattern has been found in other studies. In the study by Shelley et al. [18], nfvPPA was associated with CBS with pathological proven CBD. Hu et al. [69] found apraxia of speech in some of their CBS-CBD patients but in none of the CBS-AD patients.

On the other hand, CBS-AD subgroups had a higher proportion of lexical-semantic errors (especially word-finding difficulties) and more filled pauses in comparison to controls. Filled pauses are probably related to word-finding difficulties, which means that CBS-AD presented mainly impairments related to the lexical level. Word-finding difficulties have been largely reported in studies of connected speech of clinical AD patients [23,25,26,29,70], as well as in confrontation naming tasks [71,72,73] and in the connected speech of lvPPA patients [25,32,74]. However, our sample was reduced, and it is necessary to confirm these findings with a larger sample.

### 4.4. Limitations and Future Directions

This preliminary and exploratory study illustrates features of CBS oral discourse production, but findings need to be interpreted with caution due to some limitations. Our sample size is small, resulting in a lower statistical power of the analysis. Another limitation is that most patients were already in an advanced stage of the disease, and they could not produce a sample of 150 words, which is considered the minimum necessary for connected speech analysis [75]. Finally, we elicited oral discourse only from picture description, which puts some constraints on the production of a wide variety of syntactical structures. Therefore, our findings need to be confirmed by studies with larger samples and more variety of connected speech samples. Longitudinal studies, including patients in the initial stages of the disease, are needed to further understand how CBS impacts connected speech. These methodological improvements might shed light on differences between the phenotypes of CBS related and not related to AD.

## 5. Conclusions

In our study, CBS patients presented with a reduced lexical output, especially for open-class words, with a lower speech production rate, simple syntactic structures, word retrieval difficulties and impaired informativeness. Therefore, our first hypothesis was confirmed, as the group presented impairments related to speech, fluency, syntactic, and lexical-semantic levels. When looking at CBS patients according to the underlying pathology, CBS-non-AD presented mainly with impairments related to motor speech disorders and syntactic complexity, a speech profile similar to nfvPPA [32,39]. Instead, patients with CBS due to AD showed mainly lexical deficits, a linguistic pattern similar to lvPPA and typical clinical presentations of AD [26,40].

Despite being a preliminary study, our findings suggest that connected-speech measures are promising markers of CBS underlying pathology. Furthermore, by characterizing the discourse of CBS patients, our study contributes to refining behavioral interventions.

## Figures and Tables

**Table 1 brainsci-12-01705-t001:** Description of discursive variables.

Discursive Variable	Definition
**Speech rate and speech sound errors**	
Number of words	Total number of words produced. Contractions were considered as one word. False starts, repaired sequences, and filled pauses (see “other disruptions of fluency”) were not included in word counting
Speech production rate	Total number of words/duration of the sample without pauses
Phonological paraphasias phw	(Total number of phonological paraphasias/number of words) × 100
Distortions phw	(Number of distortions/number of words) × 100
**Other disruptions to fluency**	
False starts phw	Single words that are abandoned after some phonemes are produced. (Total number of false starts/number of words) × 100
Repaired sequences phw	Complete words (or a single complete word) that are reworked. (Total number of repaired sequences/number of words) × 100
Filled pauses phw	Examples of filled pauses are “hmmm” and “aah” (in Portuguese: “hum” and “é”). (Total number of filled pauses/number of words) × 100
Incomplete sentences phw	Sentences abandoned (and not repaired) and sentences missing obligatory elements (a verb and its obligatory arguments). (Total number of sentences abandoned/number of words) × 100
**Lexical-semantic level**	
Open-class proportion	Verbs, nouns, adjectives, and adverbs ending in -ly (in Portuguese: -mente) were considered open-class words. Light verbs (verbs with little semantic content), conjunctions, prepositions, articles and pronouns were counted as closed-class words. Open-class words/closed-class words
Verb proportion	Verbs/verbs + nouns
Lexical-semantic errors phw	(Semantic paraphasias, paraphrases, and word-finding difficulties/number of words) × 100
% CIU	Correct information units (CIU) are words that are relevant to the context (the picture) and informative. Total number of correct information units/total number of words produced
**Syntactic structure and complexity**	
Number of utterances	Total number of utterances
Number of sentences	Total number of complete sentences, i.e., a sentence with at least a verb and its obligatory arguments
Morphosyntactic errors phw	(Total number of ungrammatical sentences, inflectional errors, and absence of determiners/number of words) × 100
Embeddings phw	(Total number of sentences embedded within another sentence/number of words) × 100
Mean length of sentences	Total number of words in sentences/total number of sentences
Proportion of sentences	Total number of complete sentences/total number of utterances
Syntax production rate	Total number of words in sentences/total number of words

Abbreviation: CIU = correct information units; phw = per hundred words.

**Table 2 brainsci-12-01705-t002:** Demographic and clinical characteristics of the participants.

	CBS (*n* = 14)	CBS-Non-AD (*n* = 10)	CBS-AD (*n* = 4)	CG (*n* = 15)	CBS vs. CG *p*-Value	CBS-Non-AD vs. CBS-AD vs. CG *p*-Value
Age (y)	67.9 (8.4)	66.4 (8.3)	71.5 (8.7)	67.3 (8.1)	0.813	0.746
Education (y)	8.8 (5.8)	9.3 (5.9)	7.5 (6.0)	9.3 (5.4)	0.914	0.875
Sex (F/M)	9/5	5/5	4/0	10/5	0.892	0.203
Hand dominance (R/L)	12/2	8/2	4/0	14/1	0.500	0.430
Symptom duration (y)	4.6 (2.1)	4.7 (2.4)	4.5 (1.2)	--	--	0.770 *
CDR	1.9 (0.7)	1.9 (0.7)	2.0 (0.8)	--	--	0.941 *
FAQ	19.9 (7.8)	18.6 (7.6)	23.0 (8.4)	--	--	0.337 *
NPI	17.2 (16.7)	12.9 (8.7)	26.7 (27.1)	--	--	0.394 *
H&Y	2.9 (1.3)	3.0 (1.3)	2.7 (1.7)	--	--	0.774 *

Note: Comparison analyses across all groups were performed using ANOVA (normally distributed data) or the Kruskal–Wallis test (non-normal distribution). The CBS vs. CG and CBS-non-AD vs. CG were carried out using Student’s *t*-test (normal) or the Mann–Whitney test (non-normal). * CBS-non-AD vs. CBS-AD comparison using the *t*-test or the Mann–Whitney test. Sex and hand dominance were compared across groups using chi-squared test. Data are reported as the mean (SD). Abbreviations: CBS = corticobasal syndrome, CBS-non-AD = corticobasal syndrome not related to Alzheimer’s disease, CBS-AD = corticobasal syndrome related to Alzheimer’s disease; SD = standard deviation, y = years, CDR = Clinical Dementia Rating, FAQ = Functional Activities Questionnaire, NPI = Neuropsychiatric Inventory, and H&Y = Hoehn and Yahr scale.

**Table 3 brainsci-12-01705-t003:** Performance on cognitive and language tests across diagnostic groups.

	CBS (*n* = 14)	CBS-Non-AD (*n* = 10)	CBS-AD (*n* = 4)	CG (*n* = 15)	CBS vs. CG *p*-Value	CBS-Non-AD vs. CBS-AD vs. CG *p*-Value
ACE-R						
Total score	44.0 (19.4)	48.5 (18.4)	32.8 (19.5)	86.1 (9.0)	**<0.001**	**<0.001 ^ab^**
Attention	10.8 (3.9)	12.3 (3.0)	7.0 (3.7)	16.9 (1.3)	**<0.001**	**<0.001 ^ab^**
Memory	9.7 (6.8)	11.6 (6.7)	5.0 (5.3)	20.3 (5.3)	**<0.001**	**0.001 ^ab^**
Fluency	2.6 (2.6)	2.7 (2.7)	2.3 (2.6)	10.3 (2.2)	**<0.001**	**<0.001 ^ab^**
Language	15.3 (6.6)	15.4 (6.4)	15.0 (8.0)	24.4 (2.4)	**<0.001**	**0.003 ^a^**
Visuospatial	5.6 (3.7)	6.5 (4.0)	3.5 (0.6)	14.3 (1.7)	**<0.001**	**<0.001 ^ab^**
WAB-R						
Aphasia Quotient	83.0 (11.6)	83.8 (11.5)	81.0 (13.6)	98.2 (1.6)	**<0.001**	**<0.001 ^ab^**
Spont speech	16.8 (2.3)	16.9 (2.3)	16.5 (2.9)	19.9 (0.3)	**<0.001**	**<0.001 ^ab^**
Aud comp	8.6 (1.4)	8.6 (1.6)	8.7 (1.2)	9.9 (0.1)	**0.003**	**0.011 ^b^**
Repetition	8.7 (1.0)	8.8 (0.9)	8.5 (1.3)	9.7 (0.3)	**<0.001**	**0.002 ^ab^**
Naming	7.1 (2.3)	7.3 (2.5)	6.9 (1.8)	9.6 (0.6)	**<0.001**	**0.001 ^ab^**

Note: Comparison analysis was first performed between CBS vs. CG and then performed between CBS-non-AD vs. CBS-AD vs. CG with Student’s *t*-test/ANOVA or the Mann–Whitney/Kruskal–Wallis test. Data are reported as the mean (SD); a = CBS-non-AD vs. CG *p*-value > 0.05, b = CBS-AD vs. CG *p*-value > 0.05, c = CBS-non-AD vs. CBS-AD *p*-value > 0.05. Bold-faced values are statistically significant according to *p* values. Abbreviations: CBS = corticobasal syndrome, CBS-non-AD = corticobasal syndrome not related to Alzheimer’s disease, CBS-AD = corticobasal syndrome related to Alzheimer’s disease; ACE-R = Addenbrooke’s Cognitive Examination-revised; WAB-R = Western Aphasia Battery—Revised; SD = standard deviation, Spont speech = Spontaneous speech, and Aud comp = Auditory comprehension.

**Table 4 brainsci-12-01705-t004:** Performance on discursive measures related to “speech rate and speech sound errors” and “other disruptions to fluency” by diagnostic group.

	CBS (*n* = 14)	CBS-Non-AD (*n* = 10)	CBS-AD (*n* = 4)	CG (*n* = 15)	CBS vs. CG *p*-Value	CBS-Non-AD vs. CBS-AD vs. CG *p*-Value
Speech rate and speech sound errors						
Number of words	53.4 (22.5)	48.5 (25.1)	65.5 (6.2)	72.9 (27.3)	0.051	0.065
Speech prod rate	0.8 (0.2)	0.8 (0.2)	0.9 (0.3)	1.4 (0.3)	**<0.001**	**<0.001 ^a^**
Phon paraphasias phw	0.0 (0.0)	0.0 (0.0)	0.0 (0.0)	0.0 (0.0)	--	---
Distortions phw	4.2 (9.0)	5.4 (10.5)	1.2 (1.4)	0.0 (0.3)	0.077	0.058
Other disruptions to fluency						
False starts phw	2.1 (3.0)	1.73 (3.5)	3.0 (1.1)	1.8 (2.6)	0.847	0.109
Repaired sequences phw	8.1 (6.5)	7.5 (6.6)	9.6 (7.2)	5.5 (3.3)	0.505	0.652
Filled pauses phw	1.8 (2.0)	1.3 (2.0)	3.0 (1.7)	0.7 (1.4)	0.146	**0.024 ^b^**
Incomplete sentences phw	4.0 (6.1)	3.4 (6.7)	4.1 (4.3)	0.6 (1.2)	0.089	0.120

Note: Comparison analysis was first performed between CBS vs. CG and then performed between CBS-non-AD vs. CBS-AD vs. CG with Student’s *t*-test/ANOVA or the Mann–Whitney/Kruskal–Wallis test. Data are reported as the mean (SD); *p* is significant at the 0.05 level; a = CBS-non-AD vs. CG *p*-value > 0.05, b = CBS-AD vs. CG *p*-value > 0.05, c = CBS-non-AD vs. CBS-AD *p*-value > 0.05. Bold-faced values are statistically significant according to *p* values. Abbreviations: phw = per hundred words, CBS = corticobasal syndrome patients, CBS-non-AD = corticobasal syndrome not related to Alzheimer’s disease patients, CBS-AD = corticobasal syndrome related to Alzheimer’s disease patients; SD = standard deviation, s = seconds, speech prod rate = speech production rate, and Phon paraphasias = phonological paraphasias.

**Table 5 brainsci-12-01705-t005:** Performance on discursive measures related to “lexical-semantic level” by diagnostic group.

	CBS (*n* = 15)	CBS-Non-AD (*n* = 11)	CBS-AD (*n* = 4)	CG (*n* = 15)	CBS vs. CG *p*-Value	CBS-Non-AD vs. CBS-AD vs. CG *p*-Value
Open-class proportion	0.7 (0.1)	0.7 (0.1)	0.6 (0.1)	1.0 (0.2)	**<0.001**	**0.002 ^ab^**
Verb proportion	0.4 (0.1)	0.4 (0.1)	0.4 (0.0)	0.4 (0.0)	0.813	0.378
Lexical-semantic errors phw	4.4 (5.9)	3.7 (5.4)	6.3 (7.6)	0.7 (1.5)	**0.012**	**0.013 ^b^**
% CIU	72.4 (17.3)	73.8 (16.9)	68.9 (20.2)	84 (6.9)	**0.037**	0.101

Note: Comparison analysis was first performed between CBS vs. CG and then performed between CBS-non-AD vs. CBS-AD vs. CG with Student’s *t*-test/ANOVA or the Mann–Whitney/Kruskal–Wallis test. Data are reported as the mean (SD); *p* is significant at the 0.05 level; a = CBS-non-AD vs. CG *p*-value > 0.05, b = CBS-AD vs. CG *p*-value > 0.05, c = CBS-non-AD vs. CBS-AD *p*-value > 0.05. Bold-faced values are statistically significant according to *p* values. Abbreviations: phw = per hundred words, CIU = correct information units, CBS = corticobasal syndrome patients, CBS-non-AD = corticobasal syndrome not related to Alzheimer’s disease patients, CBS-AD = corticobasal syndrome related to Alzheimer’s disease patients, and SD = standard deviation.

**Table 6 brainsci-12-01705-t006:** Performance on discursive measures related to “syntactic structure and complexity” by diagnostic group.

	CBS (n = 15)	CBS-Non-AD (n = 11)	CBS-AD (n = 4)	CG (n = 15)	CBS vs. CG *p*-Value	CBS-Non-AD vs. CBS-AD vs. CG *p*-Value
Number of utterances	11.1 (4.2)	10.4 (4.6)	12.8 (3.0)	12.7 (4.2)	0.331	0.284
Number of sentences	8.6 (5.0)	7.9 (5.6)	10.2 (3.2)	8.8 (4.4)	0.652	0.464
Embeddings phw	0.7 (1.2)	0.1 (0.5)	2.2 (1.3)	2.4 (1.8)	**0.020**	**0.003 ^a^**
Mean length of sentences	5.6 (1.4)	5.5 (1.3)	5.7 (1.7)	6.4 (2.8)	0.172	0.376
Proportion of sentences	0.7 (0.2)	0.7 (0.2)	0.8 (0.2)	0.6 (0.2)	0.270	0.457
Syntax production rate	0.8 (0.2)	0.8 (0.1)	0.8 (0.2)	0.7 (0.2)	0.747	0.815
Morphosyntactic errors phw	0.0 (0.0)	0.0 (0.0)	0.0 (0.0)	0.0 (0.0)	0.652	0.699

Note: Comparison analysis was first performed between CBS vs. CG and then performed between CBS-non-AD vs. CBS-AD vs. CG with Student’s *t*-test/ANOVA or the Mann–Whitney/Kruskal–Wallis test. Data are reported as the mean (SD); *p* is significant at the 0.05 level, a = CBS-non-AD vs. CG *p*-value > 0.05, b = CBS-AD vs. CG *p*-value > 0.05, c = CBS-non-AD vs. CBS-AD *p*-value > 0.05. Bold-faced values are statistically significant according to *p* values. Abbreviations: phw = per hundred words, CBS = corticobasal syndrome patients, CBS-non-AD = corticobasal syndrome not related to Alzheimer’s disease patients, CBS-AD = corticobasal syndrome related to Alzheimer’s disease patients, and SD = standard deviation.

## Data Availability

Not applicable.

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
