# Peer review of "The Discourse Profile in Corticobasal Syndrome: A Comprehensive Clinical and Biomarker Approach"

_brainsci, 2022, doi:10.3390/brainsci12121705_

Round 1

Reviewer 1 Report

The article is quite interesting, wants to do a lot of things, it is dense enough, and it would be an appropriate reading for the audience of the Journal. 

The Introduction should highlight better why the paper is relevant in its field of studies, the specific research goals and how the Authors plan to achieve them. 

Where is the Literature Review? Scattered here and there, all over the paper. It would be more reasonable, and even necessary, to develop a specific section, entitled "Literature Review", after the Introduction and before the Methods, where the Authors list and discuss, briefly but systematically, all the works they have used and cited in their paper and relevant texts in the field. That would be welcomed by a specialized audience and would help a lot a non-specialized audience as well. 

Methods are ok, in my opinion, but, since I am essentially a Linguist, they need to be double-checked by other Reviewers more proficient than me in neurosciences in general. 

The results are neat, quite comprehensive, at least for a paper like this. 

The Discussion is exhaustive enough, I would still add something, at the level of analysis and comments, especially at the linguistic level (I mean, by highlighting linguistic elements not only in the 'applied way', but also by looking at the theoretical foundations), but nothing can be perfect. 

The Conclusion should be expanded, adding more 'synthesis' and also 'mirroring' the Introduction, summarizing the main research goals of the paper, explaining how they have been achieved and how this research gives a contribution to the field. 

The English language is quite clear, not native, not always 'genuine', but solid and relatively effective. It would benefit, nonetheless, from the help and a re-reading by a native speaker. 

The paper can be considered for publication, but 

- it needs a proper Literature Review; 

- it needs expansions of Introduction and Conclusion; 

- it needs a possible expansion of the Discussion at the linguistic level; 

- it needs a possible revision of the language by a native-speaker. 

Thank you very much. 

Author Response

We thank the Brain Sciences journal for the thorough evaluation of our article and thank the editor for the opportunity to revise and improve it. We have carefully considered all the suggestions received and implemented most of them. In a few cases, we present justifications for not implementing the suggestions.

We made substantial modifications in the Introduction, in order to include a comprehensive review of the literature, following suggestions of the reviewers. In addition, we have added small amendments and corrections to the text, after the language revision. All changes in the manuscript can be tracked.

We remain willing to improve the article if necessary.

Sincerely,

Isabel Junqueira de Almeida and Maria Teresa Carthery-Goulart (on behalf of co-authors)

University of São Paulo

Brazil

Reviewer #1

We thank the Reviewer #1 for all his/her suggestions and corrections. We will address them individually below:

  1. The article is quite interesting, wants to do a lot of things, it is dense enough, and it would be an appropriate reading for the audience of the Journal.  The Introduction should highlight better why the paper is relevant in its field of studies, the specific research goals and how the Authors plan to achieve them. 

Thank you for the thorough evaluation of our paper and for thoughtful suggestions. We included some lines specifying the relevance of the study to the field (lines 195-200) and the specific goals (lines 201-208).

  1. Where is the Literature Review? Scattered here and there, all over the paper. It would be more reasonable, and even necessary, to develop a specific section, entitled "Literature Review", after the Introduction and before the Methods, where the Authors list and discuss, briefly but systematically, all the works they have used and cited in their paper and relevant texts in the field. That would be welcomed by a specialized audience and would help a lot a non-specialized audience as well. 

- Thank you for your suggestion. We made substantial changes in the Introduction and included some subsections for a more detailed presentation of previous studies. There is paucity of studies focusing on connected-speech in CBS, so we included some paragraphs about previous connected speech research on different neurodegenerative disorders that can manifest clinically as CBS (lines 144-192). However, we checked the structure of papers in Brain and Sciences journal and noted that the literature review is usually presented as part of the introduction. Thus, we kept this pattern to be consistent with the journal’s layout.

  1. Methods are ok, in my opinion, but, since I am essentially a Linguist, they need to be double-checked by other Reviewers more proficient than me in neurosciences in general. 

- Thank you for your comment.

  1. The results are neat, quite comprehensive, at least for a paper like this. 

- Thank you for your comment.

  1. The Discussion is exhaustive enough, I would still add something, at the level of analysis and comments, especially at the linguistic level (I mean, by highlighting linguistic elements not only in the 'applied way', but also by looking at the theoretical foundations), but nothing can be perfect. 

- We really appreciate your suggestion, however this preliminary study focused on clinical aspects due to some methodological constraints that we describe in the Discussion. Some limitations preclude a deeper linguistic analysis with views to discuss theoretical models. Among them, the limited speech samples and a single type of elicited connected-speech. We intend to increase the sample and include patients in a mild stage of the disease. In addition, we intend to assess other types of discourse production (argumentative speech, procedural discourse etc.). We included these limitations in the Discussion along with directions for future work (line 702).

  1. The Conclusion should be expanded, adding more 'synthesis' and also 'mirroring' the Introduction, summarizing the main research goals of the paper, explaining how they have been achieved and how this research gives a contribution to the field. 

- Thank you for your suggestion. We agree that our writing was not clear, so we restructured our Discussion and Conclusions. The summary of the main research goals and synthesis of results are in the beginning of the Discussion (lines 528-540).  We added a sentence in the last paragraph of the Conclusion to explain the contribution to the field, aligned to the panorama detailed in the introduction (line 743).

  1. The English language is quite clear, not native, not always 'genuine', but solid and relatively effective. It would benefit, nonetheless, from the help and a re-reading by a native speaker. 

- Thank you for your suggestion. The paper was reviewed to achieve more clarity and accuracy of English language.

Reviewer 2 Report

The aim of the current study was characterize the oral discourse of corticobasal syndrome (CBS) patients and to verify whether measures obtained during a semi-spontaneous speech production could differentiate CBS patients from controls.

This study is interesting and a nice well-written manuscript. Nevertheless, this manuscript needs some improvements and corrections before publishing may be possible.

General points:

Please add a list of abbreviations before References section to your manuscript.

Special points:

Introduction

Lines 38-39: please add multiple references at the end of this sentence.

Lines 43-45: please describe exactly all these studies.

Lines 57-58: please add multiple references at the end of this sentence.

Line 92: please add multiple references at the end of this sentence.

Methods

Lines 130-134: Please add the exactly information about all your patients: number totally, number of female, number of male and the mean age.

Lines 150-152: please add also the exactly organisation name, date and the number of the permission for all your experiments.

Lines 154-157: please add references at the end of each these sentences.

Lines 179-180: please add multiple references at the end of this sentence.

Discussion

Lines 394-396: please add multiple references at the end of this sentence.

Author Response

We thank the Brain Sciences journal for the thorough evaluation of our article and thank the editor for the opportunity to revise and improve it. We have carefully considered all the suggestions received and implemented most of them. In a few cases, we present justifications for not implementing the suggestions.

We made substantial modifications in the Introduction, in order to include a comprehensive review of the literature, following suggestions of the reviewers. In addition, we have added small amendments and corrections to the text, after the language revision. All changes in the manuscript can be tracked.

We remain willing to improve the article if necessary.

Sincerely,

Isabel Junqueira de Almeida and Maria Teresa Carthery-Goulart (on behalf of co-authors)

University of São Paulo

Brazil

  • We thank the Reviewer #2 for all his/her suggestions and corrections. We will address them individually below:

This study is interesting and a nice well-written manuscript. Nevertheless, this manuscript needs some improvements and corrections before publishing may be possible.

General points:

Please add a list of abbreviations before References section to your manuscript.

  • We thank the reviewer for the thorough evaluation of our paper. We included a list of abbreviations before References.

Special points:

 Introduction

Lines 38-39: please add multiple references at the end of this sentence.

  • We thank the reviewer for the suggestion. We added other references (lines 39-40)

Lines 43-45: please describe exactly all these studies.

  • The studies were described in more detail in lines 60-85. Following your suggestion, we added some information about two of the studies that were cited on lines 43-45, but were not described.

Lines 57-58: please add multiple references at the end of this sentence.

  • We followed your suggestion and included multiple references (line 62).

Line 92: please add multiple references at the end of this sentence.

  • As we have restructured the Introduction, this sentence changed. However, the same idea is present in line 92, with references. Thank you for the suggestion.

Methods

Lines 130-134: Please add the exactly information about all your patients: number totally, number of female, number of male and the mean age.

  • Following your suggestion, missing information was added (line 314). Thank you.

Lines 150-152: please add also the exactly organization name, date and the number of the permission for all your experiments.

  • We added missing information regarding ethics committee (lines 339-341). Thank you.

Lines 154-157: please add references at the end of each these sentences.

  • All tests used are described and referenced in the following paragraph (lines 348-357). When a Brazilian version of the test was used, both references are given. Thank you.

Lines 179-180: please add multiple references at the end of this sentence.

  • The WAB-R reference was added (line 362)

Discussion

Lines 394-396: please add multiple references at the end of this sentence.

  • Thank you for your suggestion. There are few papers that address this issue (the reduction in the language output in CBS). We added one reference at the end of lines 396 (in the revised manuscript: line 592). Also, in the end of the paragraph, there are other references regarding this subject.

Round 2

Reviewer 1 Report

The paper has been improved, and surely can be considered for publication. 

Thank you very much. 

Reviewer 2 Report

Thank you for all corrections.